# Effect of Environmental Regulation on Industrial Solid Waste Pollution in China: From the Perspective of Formal Environmental Regulation and Informal Environmental Regulation

**DOI:** 10.3390/ijerph17217798

**Published:** 2020-10-25

**Authors:** Bei Xiong, Ruimei Wang

**Affiliations:** College of Economics and Management, China Agricultural University, Beijing 100083, China; xiongbei_2008@163.com

**Keywords:** formal environmental regulation, informal environmental regulation, industrial solid waste, panel threshold model, China

## Abstract

To address the concern of environmental pollution, it is necessary to study the effect of environmental regulation on industrial solid waste emission reduction in China. This paper aimed to analyze the effectiveness of provincial environmental regulation (both formal and informal) on the industrial solid waste emission reduction. The results show that both the effect of formal and informal environmental regulations on industrial solid waste emission intensity present an inverted “U” shape. The threshold value of per capita GDP as an indicator variable is CNY 16,299 and CNY 15,572 respectively. The effect on pollution emission reduction will appear when the value is higher than the threshold, and the two-way transmission mechanism between formal and informal environmental regulations does exist. When GDP per capita exceeds CNY 27,961, there is a phenomenon of “rebound” in the effect of informal environmental regulation on pollution reduction. Based on the findings, it was suggested that both formal and informal environmental regulation should be promoted to achieve the goal of industrial solid waste emission reduction. The coordination between formal and informal environmental regulation should be considered when the government makes policies. Different environmental regulation policies should be implemented in different regions. Informal regulation should be enriched and further promoted. Environmental law should play an important role in maintaining the public’s participation in environmental regulation to prevent the failure of informal environmental regulation.

## 1. Introduction

China is facing severe environmental problems, especially in solid waste pollution, due to the unprecedented rate of industrialization, urbanization and the continuous improvement of people’s living standards [1]. As the second largest economy, China produces the largest amount of solid waste in the world [2,3]. According to the projection of the World Bank, the total amount of solid waste in China will be over 480 million tons in 2030 [4]. China is facing great challenges all over the country, and waste incineration is going to be an important pollution source [5]. Industrial solid waste is one of the main sources of municipal solid waste in China [6]. The volume of industrial solid waste has dramatically increased. According to the China Statistical Yearbook, the amount of annual industrial solid waste increased from 1756.32 million tons in 2007 to 3315.92 million tons in 2017 [7]. The industrial solid waste in China has increased by 88.8% in 10 years, which means that the rapid growth of solid waste has become a serious problem. Currently, sanitary landfill is the dominant method for solid waste treatment, treating 57.2% of total waste [6]. Improper solid waste management can cause dioxin pollution, other environmental issues and even health hazards [8]. Solid waste pollution which affects public health and environment cannot be ignored with its increasingly serious threats. At present, the phenomenon of a “waste mountain” surrounding many large and medium-sized cities in China is relatively common. In the process of the suburbanization of a large number of urban industrial enterprises, various solid pollutants are left in the soil affect the health of residents. A large number of hazardous wastes in production and life have not been effectively and harmlessly disposed, and medical wastes are mixed into household garbage or even illegally reused. Illegal dismantling, the processing of waste materials, incineration, pickling, soil smelting and other activities exist in many places, resulting in the inability to cultivate the local soil, undrinkable water, and serious air pollution [9,10,11].

Under the pressure of severe environmental and ecological problems and their effects on public health, the Chinese government has implemented laws, regulations, and policies to control solid waste [12]. Policymakers and economists have attempted to address this issue. The studies of the effect of regulations on environmental protection are mainly carried out from three aspects: (1) Blackman et al. [13] explored inspections enforced by an environmental agency in Mexico, and find that environmental regulatory pressure is not associated with pollution reduction; (2) Lanoie et al. [14] argued that environmental regulation will increase discharge to firms and the cost of pollution control, crowding out productive resources, and reduce market competitiveness and productivity. As a result, it makes more difficult to manage environmental problems; (3) Zheng et al. [15] find that environmental regulation has significantly improved the air quality. These studies reveal that the responses of pollution emissions to external regulations remain are mixed and controversial. Concerning such relationships, there is a number of studies on environmental pollution from a national perspective in China. However, most of them are focused on the effects of regulation on air pollution [16,17] and water pollution [18,19]. Only a few studies have been involved in industrial solid waste pollution [20,21].

Though these studies provide evidence for understanding the relationship between environmental quality and regulation, there is still a number of important questions that are not been fully addressed, listed subsequently. These include:

Are China’s current environmental regulatory policies effective in reducing industrial solid waste pollution? Is there a non-linear relationship between environmental regulation and industrial solid waste emission? How do formal and informal environmental regulations exert impact on solid waste emission efficiency in different contexts? Is there an interaction mechanism between formal and informal environmental regulations to promote the efficiency of industrial solid waste emission?

To answer these questions, this paper aimed to analyze the effectiveness of provincial environmental regulation (both formal and informal) on the industrial solid waste emission reduction. For this purpose, this paper constructs an econometrics model. Variables such as industrialization, urbanization, and other factors are selected as determinants in the model, since they are considered to exert a major impact on environmental quality. This study focuses on the impact of informal regulation on pollution, particularly inspired by studies on the effect of informal regulation on pollution control. Accordingly, some proxy variables for informal regulation are selected in the model to capture the impact of informal regulation on the control of industrial solid waste pollution.

The marginal contributions of this article is: first, regarding the environmental Kuznets curve (EKC), an important supplement of the research, according to the assumption of a set, if the formal and informal environmental regulation and pollution emission is an inverted in a “U” shaped curve relationship, is in linear relationship between environmental regulation and economic growth under the premise of further confirmation per capita income and pollution degree of the inverted “U” shaped curve relationship, and similar studies will provide a practical basis for the future. Secondly, concerning informal environmental regulation, as an important link of environmental regulation to the research category, and previous research on informal environmental regulation noted considerable neglect and deficiencies, but this article will not only consider the pollution reduction effect of informal environmental regulation in a separate module analysis and on the discussion the interaction of formal environmental regulation, provide theoretical support for further analyzing informal environmental regulation factors. Third, a rich environmental regulation and pollution emission reduction-related research topics are well covered, on the one hand, the effect on China’s environmental regulation of industrial solid waste emission reduction research is still relatively lacking, and on the other hand, and we fully consider the “strong government and weak society” under the condition of informal environmental regulation failure problems, providing a research basis for the all-round implementation of pollution emission reduction policy.

The remaining of the paper is organized as follows. Section 2 summarizes the literature of environmental regulation (both formal and informal) and environmental pollution and presents the research hypothesis proposed by this paper. Section 3 describes the data, variables and the panel threshold method. Section 4 reports the results. A detailed discussion is provided in Section 5. The paper concludes with some policy implications in Section 6.

## 2. Theoretical Background and Hypotheses

### 2.1. Environmental Regulation and Environmental Pollution

Externalities mean that it is difficult to solve the problem of environmental pollution only by relying on market forces. This requires environmental regulation to intervene in the environmental behavior of producers or consumers. Pigou proposed the “Pigou tax” in 1932, setting a precedent for environmental regulation. Subsequently, Coase criticized Pigou’s methods to correct externalities on the grounds that the “Pigou tax” limited economic choices, emphasizing the important role of property rights and property right transactions in environmental regulation. New institutional economics regards the institution as a social game rule composed of formal rules and informal rules; accordingly, environmental regulations can be divided into formal and informal environmental regulations. It involves various stakeholders, which includes the government, the environmental protection department, the public and non-governmental organizations (NGOs), and so on. Formal environmental regulation mainly comes from the government and the environmental protection department, and informal environmental regulation mainly comes from the public and environmental NGOs [22].

There are more and more studies on the effect of environmental regulation on environmental pollution. Many related studies are based on traditional theories to study the relationship between environmental quality and economic development [23]. Foreign scholars investigated environmental regulations earlier. However, most literature is theoretical instead of empirical. In general, these studies mainly focus on the following two aspects: (1) the impact of environmental regulation on environmental pollution combined with the environmental Kuznets curve (EKC) [23]; and (2) the effect of environmental regulation on pollution emissions at the enterprise level.

The EKC attempts to reveal the long-term curve relationship between environmental pollution and economic growth, which has aroused heated discussions in the field of environmental regulation. According to recent studies, Zhang et al. (2009) [21] verified the existence of EKC, and concluded that a systemic policy of environmental regulation can change the figure of the EKC curve. Due to heterogeneity between different regions, EKC behaves differently in different regions. Gao et al. (2011) [24] found that there was an inverted U-shaped EKC in eastern regions and U-shaped EKC in west regions, but no EKC in central regions. Change et al. (2017) [20] found that the effect of formal and informal environmental regulations on the pollution emission intensity of China during the period 2001–2014 presents an inverted “U” shape.

Based on the above literature, the effect of environmental regulation on emission reduction needs further research. To verify the relationship between environmental regulation and solid waste pollution, to enrich the research on the Kuznets curve of the solid waste, this paper puts forward the following hypothesis:

**Hypothesis** **1** **(H1):***The relationship between environmental regulation and solid waste pollution meets the Kuznets curve, and there is an inverted U-shaped relationship between formal and informal environmental regulations and pollution level*.

### 2.2. Formal Environmental Regulation and Informal Environmental Regulation

Formal environmental regulation usually refers to the various mechanisms implemented by government agencies to control pollution emissions. In general, formal regulation combines a pollution emission standard system with a sanctions regime for non-compliance. Incentive-based measures, such as taxes on energy input or the output of production, are usually the responsibility of public authorities. Therefore, they are also formal environmental regulations [25].

Initially, environmental regulations were regarded as the government’s mandatory policies and regulations to intervene in the use of environmental resources. With the attention to environmental problems and the development of regulatory tools, environmental regulation also includes incentive regulation and voluntary regulation. For example, environmental taxes, input and output taxes [25], pollution control subsidies [26] and tradable sewage permits [27] are incentive environmental regulations.

Numerous studies have shown that there is a positive correlation between formal environmental regulation and environmental performance (see Table 1). Formal regulation (enforcement of standards and monitoring of emissions) are generally considered as key determinants of environmental performance. Although conventional wisdom holds that formal regulatory pressures are relatively low, the environmental performance of developing countries has been driven by such pressures [25].

Afterwards, informal environmental regulations extend the connotation of environmental regulations. There are relatively few studies on informal environmental regulation compared with that on formal environmental regulation. Pargal and Wheeler [28] first came up with the concept of informal environmental regulation. The importance of informal regulation in environmental protection has been well recognized by scholars. When formal environmental regulation is weak or missing in developing countries, many agencies will make self-interested agreements with local polluters to reduce their emissions, which is informal environmental regulation. Kathuria [29] believes that formal environmental regulations have certain limitations in pollution control in developing countries due to their asymmetric information. She emphasizes the importance of informal environmental regulation in achieving environmental goals.

As Blackman [30] points out, there are spillover effects and feedback effects between formal and informal environmental regulation mechanisms, particularly in developing countries. Pargal et al. [31] firstly made such an argument that that formal rules and regulations, especially the supervision and implementation of standards, often reflect the bargaining power of local communities and their implementation is not uniform. Cole et al. [32] mention that if the community lobbies local authorities that regulate firms, there can be an indirect impact of informal regulations. Kathuria [29] recently made a similar point, pointing out that one of the reasons companies may respond to informal rules is that when the environment is underperforming, formal rules tend to become stronger.

Table 1 summarizes previous studies on the effect of environmental regulation on pollution control. Two sets of major variables were identified. At present, domestic and foreign studies on the intensity of formal environmental regulation can be divided into two categories: regulatory behavior and regulatory effect. Regulatory behavior variables refer to the severity of pollution standards set by the government, as well as the implementation and supervision of environmental regulations. The main indicators include: (1) the number of environmental standards or regulations issued by local governments; (2) expenditure on pollution control and supervision, and the frequency of law enforcement; (3) environmental supervision, etc. The regulation effect reflects the governance effort by examining the actual effect of pollution control, mainly including pollution intensity, expressed by pollutant emission per unit of industrial added value [33]. Informal regulation takes many forms. In the existing research, the informal environmental regulation intensity is mainly measured by compensation by social ostracism of the polluting firm’s employees, community groups, the threat of physical violence, and efforts to monitor and publicize the firm’s emissions [31], literacy rate and the public opinion support rate in parliamentary elections [34], social reputation [35], media exposure to pollution events [29], public environmental awareness [36], and other indicators.

Existing research shows that there are few studies on environmental pollution from a national perspective in China, which mainly focuses on the effects of environmental regulation on water pollution and air pollution. Research on solid waste is relatively inadequate in this area. Moreover, studies on the pollution reduction effect of informal environmental regulations are seriously inadequate. Except for a few studies on informal environmental regulation in China, other studies related to environmental regulation mainly focus on formal environmental regulations, and it is even more difficult to discuss the interaction mechanism between formal and informal environmental regulations. To further verify the relationship between environmental regulation and solid waste pollution, enriching the research results of pollution reduction effect of informal environmental regulation, and exploring the two-way transmission mechanism of formal and informal environmental regulation, this paper proposes the following hypothesis:

**Hypothesis** **2** **(H2):***Regarding the effect on solid waste emissions, there is a two-way transmission mechanism between formal and informal environmental regulation, which can affect each other’s effect on solid waste emissions*.

## 3. Materials and Methods

### 3.1. Data and Variables

#### 3.1.1. Data

In consideration of the integrity and availability of data, this paper adopted a panel data of 30 provinces (excluding Tibet) in China from the year of 2003 to 2017. The data were collected from the China Statistical Yearbook, China Environment Yearbook, China Civil Affairs Statistical Yearbook, and the China Labour Statistical Yearbook. This paper quotes all monetary quantities, calculates the price of consumer price index, GDP index and commodity retail price index relative to a fixed base period of 2003. All variables are in the form of annual variables and logarithms.

#### 3.1.2. Variables

In this paper, the model takes formal and informal environmental regulations as core explanatory variables, and other influencing factors of environmental quality as control variables. To meet the needs of research design and data availability, this paper selects indicators based on existing studies. According to Change et al. (2017) [20], the multi-index form will lead to the neglect of the nonlinear relationship between the indicators in the threshold model test. The entropy weight method (EWM) is adopted in this paper, and the dependent variable and independent variable are treated as a single index form, which overcomes the problem. The entropy weight method was employed to calculate the weight of selected index variables, and the corresponding weight of each indicator variable was shown in Table 2.

(1) Dependent Variable

The measurement of regional solid waste emission includes two dimensions: total emission and emission performance. At present, China is still under the acceleration of its industrialization and urbanization, which means that it is difficult to reduce the total emissions in a short time. It is more consistent with China’s current status to measure the emission level of solid waste from the perspective of emission performance. Therefore, solid waste emission intensity is selected as the measurement index of emission reduction effect in this paper, denoted as the industrial solid waste emission intensity (ISEI), which is measured by the ratio of regional GDP to industrial solid waste emissions.

(2) Independent Variable

This study measures the intensity of environmental regulations by formal and informal environmental regulations.

Formal environmental regulations: this study uses the regulation effect to describe the intensity of formal institution. This paper argues that the pollution removal rate can not only overcome the shortcoming of a single index, but also accurately reflect the situation of environmental regulation, on the premise that specific environmental regulation policies in different regions cannot be counted. Formal environmental regulation is measured by the comprehensive utilization rate of industrial solid waste.

Informal environmental regulation: defining and measuring informal regulation is a key issue in this study. In this study, the comprehensive index method is adopted to construct the informal environmental regulation intensity, which can overcome the problems of one-sidedness and data availability that may exist in a single index. Informal environmental regulation stems from social pressure. Both the public’s attention to environmental pollution and the public’s transparency of environmental pollution disclosure have brought enormous pressure to the government and polluters, which has evolved into informal environmental regulation. From the existing research, informal environmental regulation is actually the concentrated embodiment of environmental protection consciousness. Zhao et al. [36] argue that the aim of environmental regulation is to protect the environment, which can be implemented in a tangible system or intangible consciousness. Thus, environmental regulations can be classified into explicit and implicit environmental regulations. Accordingly, implicit environmental regulations include internal and intangible concepts of environmental protection, environmental protection thinking, environmental awareness, environmental cognition, etc. The connotation of informal environmental regulation means the public awareness of environmental protection in this paper. Referring to the method proposed by Pargal and Wheeler [28], the education degree and income level are selected to measure the intensity of informal environmental regulations in this paper. Research shows that there is a positive correlation between the level of education and income and the public’s awareness of environmental protection as well as the level of public participation in environmental protection. Meanwhile, this study believes that groups with higher environmental awareness show a higher willingness to protect the environment. On the other hand, as environmental awareness does not necessarily translate into environmental behavior, so only by expressing their environmental preferences through certain channels can the public have an impact on the government’s environmental policies. Studies have shown that environmental non-governmental organizations (ENGOs) play a significant and active role in China’s urban environmental governance [38]. Thus, we choose the environmental NGO scale as an indicator of the participation channel.

(3) Control Variable

The model selects other factors affecting environmental quality as control variables.

Fiscal decentralization: the existing literature shows that fiscal decentralization not only promotes the rapid economic growth in China, but also aggravates the level of environmental pollution, which is an important factor that affects environmental pollution [39]. This study selects the ratio of revenue in local budgets to GDP to measure fiscal decentralization.

Industrial structure: industrial structure have a direct impact on the general allocation of resources and the types and quantities of pollutants, which are closely linked with the quality of production and living environment [40]. Since the main source of pollution is from the secondary industry, this study selects the proportion of the second industry to GDP to measure the industrial structure (IS).

Urbanization rate: in the process of urbanization, a large amount of infrastructure construction and the increase in consumption will undoubtedly increase energy use and solid waste emission. The urbanization rate is expressed by the ratio of the urban population in each region to the permanent resident population at the end of the year.

Technological innovation: technological innovation is one key to overcoming environmental constraints to achieve sustainable development. Through improving the efficiency of resource utilization, technological innovation can help save resources. In addition, technological innovation can also decrease pollution emissions by driving the development of environmental protection technologies [41]. This study selects the ratio of R&D expenditure to GDP to measure technological innovation.

Economic externality: as a dynamic system, the economic system needs to constantly exchange information and materials with the outside world. The relationship between FDI and environmental quality is mainly derived from the “pollution paradise hypothesis”. The more open the economy and the larger the FDI, the more likely the regions are becoming polluted havens due to the transfer of high-pollution industries. Many studies have examined the relationship between economic openness and environmental pollution [42,43]. This study selects the proportion of foreign direct investment in GDP to measure economic externality.

### 3.2. Methods

The core idea of the nonlinear threshold model is to investigate whether the correlation between explanatory variables and explained variables changes with the change of threshold variables. In other words, when the value of the threshold variable exceeds a certain critical value, the influence of an explanatory variable on the explained variable changes significantly. The above theoretical research shows that there is a nonlinear relationship between environmental regulation and pollution emission. The influence direction and degree of environmental regulation on industrial solid waste emissions may change with the economic and social changes, showing a dynamic nonlinear characteristic. In order to verify the existence of these “thresholds”, this study employed a panel threshold method to test the threshold effect and the hypotheses. The threshold regression method was developed for non-dynamic panels with individual-specific fixed effects [44]. The advantage of the threshold model is that it can endogenously divide the interval of the curve and find one or more threshold values corresponding to the indicator variables. Before the specific threshold number is determined, it is necessary to construct a multi-threshold panel model test. The regression equation is written as:(1)$$ln{\mathrm{ISEI}}_{\mathrm{it}}=a+\;\overset{m}{\underset{i=1}{\sum }}\alpha_{i}X_{\mathrm{it}}{+\eta }_{i}{\mathrm{lnfor}}_{\mathrm{it}}I\left({\mathrm{pergdp}}_{\mathrm{it}}{<\rho }_{1}\right){+\ldots +\eta }_{n}{\mathrm{lnfor}}_{\mathrm{it}}I\left(\rho_{n-1}\leq {\mathrm{pergdp}}_{\mathrm{it}}{<\rho }_{n}\right){+\mu }_{i}{+\varepsilon }_{\mathrm{it}}$$
(2)$$ln{\mathrm{ISEI}}_{\mathrm{it}}=a+\;\overset{m}{\underset{i=1}{\sum }}\alpha_{i}X_{\mathrm{it}}{+\eta }_{i}ln{\mathrm{infor}}_{\mathrm{it}}I\left({\mathrm{pergdp}}_{\mathrm{it}}{\text{<}\rho }_{1}\right){+\ldots +\eta }_{n}{\mathrm{lninfor}}_{\mathrm{it}}I\left(\rho_{n-1}\leq {\mathrm{pergdp}}_{\mathrm{it}}{<\rho }_{n}\right){+\mu }_{i}{+\varepsilon }_{\mathrm{it}}$$
where ISEI_it_ denotes the industrial solid waste emission intensity for province i in year t; a is a constant term; X_it_ denotes the control variable group; pergdp_it_ refers to GDP per_capita in different regions, which is the threshold variable affecting environmental regulation; for_it_ and infor_it_ represent for formal regulation and informal regulation; I(·) are the functions that take the value of 0 or 1; η denotes the extents of the impact of for_it_ or infor_it_ on ISEI_it_; $\mu_{i}$ measures the individual effect or heterogeneous effect; and $\varepsilon_{\mathrm{it}}$ is the error term. According to Hansen’s “threshold regression” model [44], Bootstrap’s “self-sampling” method is used to simulate the asymptotic distribution of F statistics, and the associated probability *p*-value and confidence interval are obtained.

## 4. Results

### 4.1. Panel Threshold Regression Analysis

The descriptive statistics of all variables are shown in Table 3. Before conducting the regression, we winsorize all of the continuous variables at the 1st and the 99th percentile to remove the effect of outliers. The Stata14.0 software is used for empirical analysis.

Table 4 reports the threshold effect test results of Bootstrap after sampling samples for 300 times by the “self-sampling” method. The results are as follows: the F values of the single threshold, double threshold, and triple threshold tests of formal environmental regulation are 89.63, 30.54, and 36.51, respectively, among which only the single threshold test is higher than the 95% significance level. It can be concluded that the single threshold value is that the GDP per capita is CNY 16,299. Similarly, the F value of the threshold test of informal environmental regulation is 81.21, 33.78, and 19.18, respectively. The pollution reduction effect is more sensitive to the degree of economic development, and there is a double threshold at the 95% confidence level. The first threshold is that the GDP per capita is equal to CNY 15,572, and the second threshold is CNY 27,961.

Table 5 reports the estimation results of the formal and informal environmental regulation threshold model. The regression passed the F test and Hausman test, and the threshold regression chose the fixed effects panel model. Specifically speaking, firstly, the hypothesis that the effect of formal and informal environmental regulations on pollution emission intensity presents an inverted “U” shape is established, and the existence of the environmental Kuznets curve (EKC) is proven from the side. When the GDP per capita reaches CNY 16,299, the increase in unit formal environmental regulation intensity does not cause the change of pollution emission intensity, and the pollution emission reduction effect begins to increase with the increase in formal environmental regulation intensity. When the GDP per capita is CNY 15,572, the marginal effect of informal environmental regulation on pollution emission intensity is zero, and the pollution emission reduction effect starts to increase from zero. By the end of 2013, the GDP per capita of each region had exceeded the maximum threshold value of CNY 16,299. All regions entered the stage of increasing pollution reduction effect. Due to the unbalanced development path adopted in China, the resource endowment and environmental factors in different regions were obviously differentiated, and the impact of environmental regulation was also different. Generally, the eastern regions crossed the inflection point earlier than the central and the western regions. 

Secondly, as formal environmental regulation has stronger legal effect, including administrative means, and compulsory measures, has a stronger legal effect than informal environmental regulation, the pollution reduction effect of informal environmental regulation is lower than that of formal environmental regulation [45]. In the results of Table 5, whether to the left or right of the inverted “U” curve, the effect of formal environmental regulation on the pollution emission intensity is greater than that of informal environmental regulation, and the impact on pollution emission reduction is more significant. After the appearance of the pollution reduction effect(on the right side of the inverted “U” curve), the absolute value of the pollution reduction effect coefficient of formal environmental regulation is 0.129, which is greater than that of informal environmental regulation, which is 0.068. That is to say, in the case of unit intensity change, the change degree of the pollution reduction effect caused by formal environmental regulation intensity is greater than that caused by informal environmental regulation intensity.

Finally, the threshold value of the informal environmental regulation is CNY 15,572, which is less than the threshold value of formal environmental regulation(CNY 16,299), indicating that the degree of tolerance of informal environmental regulation is lower than that of formal environmental regulation. On the one hand, informal environmental regulation represents the personal experience of ordinary people. Compared with formal environmental regulation, it basically does not include administrative procedures, and the action organization is faster. Formal environmental regulation policies need to take into account issues such as people’s livelihood, employment, local economy and performance evaluation, etc., and they will also adopt a “wait for the moment” attitude on pollution emissions.

The interaction of a pollution reduction effect of formal and informal environmental regulation has rarely been studied before. In order to verify the existence of hypothesis 2, referring to the research of Rui et al. [46], the interaction variable (*actint*) of formal and informal environmental regulations is introduced on the basis of the original threshold regression model. In order to ensure the comparability of regression results, the control variables of the new model are the same as those in Table 5 (the regression results are omitted), and the original threshold value is used for regression, as shown in Table 6.

As can be seen from the results in Table 6, firstly, in the threshold regression model with interaction terms, coefficient symbols on both sides of the threshold value of formal and informal environmental regulations are different, indicating that the impact of environmental regulations on pollution emission intensity still presents an inverted “U” curve.

From the significance of the interaction variables, as shown in hypothesis 2, there is indeed a two-way transmission mechanism between formal and informal environmental regulations. Further analysis found that when there was no informal environmental regulation factor, the absolute value of the coefficient of the effect of formal environmental regulation on pollution emission intensity in Table 6 was less than that in Table 5. This deviation reflected in the absolute value of the coefficient indicates that there is a common influence of informal environmental regulation factors in the pollution emission reduction effect of formal environmental regulation. In other words, without the informal environmental regulation function of the public, social organizations, and government supervision agencies, the pollution reduction effect of formal environmental regulation will be “compromised”. Similarly, the emission reduction effect of informal environmental regulation also has such an influence. In conclusion, the functional mechanisms of formal and informal environmental regulation are complementary and mutually reinforcing.

In addition, when the GDP per capita exceeds CNY 27,961, formal environmental regulations no longer promote the pollution reduction effect of informal environmental regulations, which can be regarded as the “rebound” phenomenon of the pollution reduction effect of informal environmental regulations. The high efficiency of formal environmental regulation leads to the “invalidity” of informal environmental regulation. The “rebound” of the pollution reduction effect is essentially the failure of informal environmental regulation. Firstly, in the case of the increase in total pollution emissions, the reduction of its emission intensity gives the public, social organizations, and government supervision agencies the illusion of environmental quality improvement, and the demand for the further improvement of the ecological environment keeps decreasing. Secondly, the effect of formal environmental regulation on pollution reduction obviously exceeds that of informal environmental regulation. Finally, under the background of social division of labor, the degree of vertical and vertical specialization keeps increasing. Formal environmental regulation makes pollution emission control more accurate and efficient, and most of the public’s participation is reduced, resulting in the “rebound” of pollution emission reduction effect of informal environmental regulation.

### 4.2. Robustness Test

In order to test the robustness of the relationship between environmental regulation and industrial solid waste emission intensity, the incremental data of industrial solid waste emission intensity in various provinces were selected as explanatory variables. Simple regression analysis was performed with the control variables unchanged. The results show that the increment of industrial solid waste emission intensity was negatively correlated with the formal environmental regulation, and after the formal environmental regulation reaches a certain intensity, the emission intensity of industrial solid waste shows a negative growth phenomenon. The linear relationship between the informal environmental regulation and the increment of industrial solid waste emission intensity is significant, as shown in Table 7.

Considering that environmental regulation has a certain lag in regulating the emission intensity of industrial solid waste, this paper delimits the lag period of environmental regulation as one year, and verifies it with the emission intensity of industrial solid waste from 2004 to 2017 and the environmental regulation data from 2003 to 2016. The results show that the result of the original threshold environmental regulation is stable. The direction of the regression coefficient is the same, and the regression coefficient is greater than the original regression result. This shows that the role of formal and informal environmental regulation does exist in hysteresis. The results will not be repeated.

## 5. Discussion

This study makes important contributions to the research on the relationship between environmental regulation and pollution emission. As far as we know, this is the first empirical study in this field specifically focusing on China’s solid waste at the macro level. The importance of this study lies in China’s current severe solid waste pollution situation and China’s emphasis on solid waste management. This study confirms the nonlinear relationship between environmental regulation and China’s industrial solid waste emissions, and decomposed environmental regulation into formal and informal regulations, confirming the interaction between the two and the difference in their effects on China’s industrial solid waste emissions. The government and the public are the subjects of two kinds of environmental regulation. The significant implication of this study is that with the help of the government and the market, a series of policies, regulations and systems for environmental governance should be introduced to improve the intensity of environmental regulation and achieve the goal of solid waste emission reduction. This study supports the importance of adopting different environmental regulation means to solve solid waste pollution and the need for different environmental regulation policies to promote sustainable development in different regions.

The research on the relationship between environmental regulation and environmental pollution is a popular topic. With the understanding of the connotation of environmental regulation, scholars gradually distinguish between formal environmental regulation and informal environmental regulation, and decompose the dimensions of these two types of regulations. To sum up, most scholars’ studies show that formal environmental regulation has a relatively obvious effect on pollution emission reduction. However, there are still differences in the existing literature on the effect of informal environmental regulation. In China’s macro-level empirical research, scholars mostly focus on air and water pollution, or integrate solid waste into the “three wastes”, rather than focusing on industrial solid waste alone. Some existing studies have used threshold regression model to test the threshold effect for “three wastes” comprehensive pollution emission or wastewater, and obtained a threshold value, but have not tested the threshold effect for solid waste. Research at the provincial level, such as the study of Change et al.(2017) [20], selected the “three wastes” as pollutants, confirming that the effect of formal and informal environmental regulations on pollution emission intensity presents an inverted “U” shape. Qingmin et al. (2019) [47] selected industrial water as the research object, and confirmed that the relationship between formal and informal environmental regulation and industrial water consumption showed an inverted U-shaped curve, and the Kuznets curve of reservoir exists. In addition, there is a two-way conduction path between the formal environmental regulation and informal environmental regulation to promote each other’s effect on industrial water use. Research at the regional level, such as the study of Qiang (2018) [48], which selects “three wastes” as the research object and the Yangtze river economic belt as the study area. The study confirms that formal and informal environmental regulation can lower the pollution level of the Yangtze river basin, and environmental regulation has regional heterogeneity on the impact of environmental pollution. Our study advances the current research by focusing on industrial solid waste as the research object at the provincial level, dividing environmental regulation into formal and informal regulation, and studying the interaction between these two types of regulation. This enriches the current empirical research on the relationship between environmental regulation and environmental pollution in China.

The limitations of this study include: firstly, although scholars have defined the indicators of formal and informal environmental regulatory intensity from different perspectives, there are still disputes among them. The indicators selected in this study are only based on existing studies, and the comprehensiveness and innovation of the indicators need to be improved. Secondly, in terms of research content, as solid waste contains industrial and domestic solid waste, this study only focuses on industrial solid waste, and the comparative study on the differences of environmental regulation effects between them needs to be considered in the future research. Thirdly, in terms of regional differences in environmental regulation, it is necessary to further study the regulatory effects and threshold differences in different regions and analyze the causes in depth. Further research can be carried out based on the above three research limitations.

## 6. Conclusions

This study was conducted to assess the impact of environmental regulation (both formal and informal) on industrial solid waste emission reduction in China. The results of the study show that the effect of formal and informal environmental regulations on pollution emission intensity presents an inverted “U” shape. The threshold value of GDP per capita as an indicator variable is CNY 16,299 and CNY 15,572, respectively. The effect of formal environmental regulation on pollution emission intensity is greater than that of informal environmental regulation. After the appearance of the pollution reduction effect, the absolute value of the pollution reduction effect coefficient of formal environmental regulation (0.129) is greater than that of informal environmental regulation (0.068). There is a bidirectional transmission mechanism between the formal and informal environmental regulation. They complement and promote each other. In different stages of industrial solid waste emission intensity, the degree of mutual influence between them is different. There is a phenomenon of “rebound” in the effect of informal environmental regulation on pollution reduction. When the GDP per capita exceeds CNY 27,961, formal environmental regulation and informal environmental regulation will no longer work together to reduce pollution. By the end of 2013, the GDP per capita of each region had exceeded the maximum threshold value of CNY 16,299, which means all regions began to show enhanced effect of pollution reduction. Environmental regulation has a greater impact on pollution emission in eastern regions.

The research conclusions can provide important policy implications for environmental governance.

(1) Both formal and informal environmental regulation should be promoted to maintain economic prosperity without harming the environment. In the process of vigorously implementing supply-side structural reform measures, we should actively play the role of environmental regulation tools in promoting pollution reduction, so as to eliminate the production capacity with low resource utilization and serious environmental pollution emissions. Increasing support for a series of technological innovation-oriented enterprises that are clean, environmentally friendly, and intelligent should be done. Informal regulation should be enriched and further promoted. Environmental education can contribute to informal regulation to a great extent. Therefore, education on the environment should be supported to foster better environmental awareness, attitude, activities, and commitment.

(2) Regarding the differences in development levels and factor endowments between regions, the government should actively guide the development of the central and western regions. Relevant policies should be inclined to the central and western regions, and measures should be taken according to local conditions. Different environmental regulation policies should be implemented in different regions to guide the sustainable development of each region.

(3) The coordination between formal and informal environmental regulation should be considered when the government makes policies. Formal and informal environmental regulations are inseparable, so how to combine different types of regulation means is more urgent than “which regulation means is the best”. In the stage of the pollution reduction effect of environmental regulation, it is necessary to change the single-center regulation mode of government as the absolute control core. To meet the needs of regulation and practice on pollution reduction, we should give full play to the supervision advantages of the market, environmental protection organizations and the public, and improve the protection of rights and interests of environmental protection laws. We should allow the public to claim civil compensation for individuals and enterprises that cause environmental pollution and eliminate the idea of a “strong government, weak society”. Informal approaches should also be noticed, designed, and applied in line with different types of formal and informal instruments to fuel economic growth in a green way.

(4) The government should continue strengthening the degree of attention to the informal environmental regulation and protecting the public, social organizations and government supervision mechanism of environmental regulation of the right to know and participate. Efforts should be made to establish a mechanism of communication and feedback channel, so as to avoid information asymmetry of informal environmental regulation failure problem.

## Figures and Tables

**Table 1 ijerph-17-07798-t001:** Previous research on the effect of environmental regulation on pollution control.

Study	Region	Data	Independent variables	Dependent variables	Method	Result
Chen et al. (2018) [23]	China	Years 1998-2012; 30 provinces	Formal regulation (proportion of environmental pollution regulation investment in GDP)	Carbon dioxide emission, soot and dust emissions, wastewater emissions	GMM	Environmental regulation is positively related to China’s environmental pollution
Costa-CampiGarcia-Quevedo et al. (2017) [27]	Spain	Years 2008-2013; 22 manufacturing sectors	Formal regulation (pollution taxes); informal regulation (self-regulation, ownership of an approved ISO 14001)	Emissions of carbon dioxide by industrial sector	Random effects model, IV method	Environmental taxes have a positive impact on pollution control; Self-regulation is essential to achieving the objectives of environmental control
Zwickl et al. (2014) [37]	America	Years 2006-2010; 68512 block groups in the 48 contiguous states	Informal regulation (neighborhood income inequality)	Industrial air pollution	Environmental Protection Agency (EPA)’s Risk Screening Environmental Indicators model, Ordinary Least Square (OLS), spatial error models	Neighborhood Gini coefficient, or the Q4/Q2 ratio, are associated with higher pollution exposure
Féres et al. (2012) [25]	Brazil	Years 1997-1999; 404 industrial plants	Formal regulation (inspections or sanctions); informal regulation (community pressure)	Technical characteristics of the firm (size, location, vintage, etc.)	A simultaneous equation model estimated with three-stage least squares (3SLS)	Pollution emissions are affected by environmental regulation (either formal or informal).; formal regulation is largely influenced by informal regulation;
Goldar et al. (2004) [34]	India	Years 1995-1999; 106 monitoring points on 10 rivers	Informal regulation (poll percentage)	Water quality	Ordered Probit model	A significant positive relationship is found between poll percentage and water quality
Wheeler et al. (1999) [28]	Indonesia and the United States	Years 1990; 2492 plants from Indonesia and the United States	Informal regulation (income per capita, percentage of population with greater than primary education, population density)	Industrial toxics, air pollution, water pollution	A model of equilibrium pollution	The article confirms the existence of significant informal regulation in both Indonesia and the United States

**Table 2 ijerph-17-07798-t002:** Explanation of variables.

Variable			Weight	Measurement
Dependent Variables	Industrial solid waste emission intensity (ISEI)		/	The ratio of industrial solid waste emissions to regional GDP
Independent Variables	Formal regulation (FR)	Industrial solid waste regulation intensity (ISRI)	/	Comprehensive utilization rate of industrial solid waste
	Informal regulation (IR)	Income level (IL)	0.2465	Average salary per employee
		Education level (EL)	0.2690	The proportion of employees with college degree or above
		Environmental NGO scale (ENGO)	0.4845	The number of environmental NGOs per 10,000 people
Control Variables	Fiscal decentralization (FD)		/	The ratio of revenue in local budgets to GDP
	Industry structure (IS)		/	The ratio of the secondary industry to GDP
	Urbanization rate (UR)		/	The ratio of non-agricultural population to total urban population
	Technological innovation (TI)		/	R&D intensity (the ratio of R&D expenditure to GDP)
	Economic externality (EE)		/	The proportion of foreign direct investment in GDP

**Table 3 ijerph-17-07798-t003:** Variable descriptive statistical analysis.

Variable			Sample Size	Mean	Sta. Err	Min	Max
Dependent Variables	Industrial solid waste emission intensity (ISEI)		450	82.04	88.94	2.25	719.40
Independent Variables	Formal regulation (FR)	Industrial solid waste regulation intensity (ISRI)	450	66.09	20.32	20.28	103.99
	Informal regulation (IR)	Income level (IL)	450	38,429.56	21,384.43	10,382.00	131,700.00
		Education level (EL)	450	13.06	9.02	2.00	55.87
		Environmental NGO scale (ENGO)	450	220.07	201.17	10.00	1022.00
Control Variables	Fiscal decentralization (FD)		450	9.70	3.22	4.73	22.73
	Industry structure (IS)		450	46.59	8.07	19.01	66.42
	Urbanization rate (UR)		450	44.25	17.70	15.58	89.60
	Technological innovation (TI)		450	1.39	1.05	0.20	6.01
	Economic externality (EE)		450	5.80	7.08	0.66	76.93

**Table 4 ijerph-17-07798-t004:** Formal and informal environmental regulation threshold effect test.

**Threshold Variable** **:** **pergdp**											
**Formal Environmental Regulation**						**Informal Environmental Regulation**					
**Threshold Type**	**F Value**	***p*** **-Value**	**Crit10%**	**Crit5%**	**Crit1%**	**Threshold Type**	**F Value**	***p*** **Value**	**Crit10%**	**Crit5%**	**Crit1%**
single	89.63	0.0100	42.56	47.35	77.73	single	81.21	0.0000	37.17	40.67	58.65
double	30.54	0.1200	33.37	37.95	43.31	double	33.78	0.0500	27.80	32.67	50.22
triple	36.51	0.2700	65.85	81.62	93.78	triple	19.18	0.6300	50.62	58.76	73.17
**Threshold Type**	**Threshold Value**	**Confidence Interval**				**Threshold Type**	**Threshold Value**	**Confidence Interval**			
		**Lower limit**		**Upper limit**				**Lower limit**		**Upper limit**	
single	1.6299	1.3769		1.6346		single	1.5572	1.5488		1.5604	
double	2.1878	2.0627		2.1879		double	2.7961	2.7398		2.8145	
triple	0.6962	0.6676		0.7116		triple	0.9690	0.9314		0.9774	

**Table 5 ijerph-17-07798-t005:** Estimation results of the formal and informal environmental regulation threshold model.

Dependent Variable: Industrial Solid Waste Emission Intensity (ISEI)			
Formal Environmental Regulation		Informal Environmental Regulation	
Variable	Fe	Variable	Fe
lnFD	−0.096 *** (−0.19)	lnFD	−0.034 *** (−0.12)
lnIS	0.221 (0.20)	lnIS	−0.395 ** (−0.19)
lnUR	−0.244 *** (−0.11)	lnUR	−0.414 *** (−0.12)
lnTI	−0.558 *** (−0.13)	lnTI	−0.496 *** (−0.13)
lnEE	−0.108 ** (−0.06)	lnEE	0.009 (0.06)
lnfor (pergdp ≤ 1.6299)	0.175 *** (0.12)	lninfor (pergdp ≤ 1.5572)	0.091 *** (0.11)
lnfor (pergdp > 1.6299)	−0.129 *** (−0.13)	lninfor (1.5572 < pergdp ≤ 2.7961)	−0.068 ** (−0.10)
\	\	lninfor (pergdp > 2.7961)	0.110 ** (0.09)
cons	3.412*** (1.03)	cons	2.205 ** (0.97)
R-square	0.6487	R-square	0.6546
N (number of observed values)	450	N (number of observed values)	450
F	64.52	F	77.72

Note: *** and ** denote statistical significance at 1% and 5% level respectively; Fe = Fixed effects; FD = Fiscal Decentralization; IS = Industry Structure; UR = Urbanization Rate; TI = Technological Innovation; EE = Economic Externality.

**Table 6 ijerph-17-07798-t006:** Estimation results of the threshold model for adding interaction items.

Dependent Variable: Industrial Solid Waste Emission Intensity (ISEI)			
Formal Environmental Regulation		Informal Environmental Regulation	
Variable	Fe	Variable	Fe
actint	−0.045 ** (−0.02)	actint	−0.014 ** (−0.03)
lnfor (pergdp ≤ 1.6299)	0.145 *** (0.12)	lninfor (pergdp ≤ 1.5572)	0.085 *** (0.17)
lnfor (pergdp > 1.6299)	−0.116 *** (−0.11)	lninfor (1.5572 < pergdp ≤ 2.7961)	−0.058 ** (−0.15)
\	\	lninfor (pergdp > 2.7961)	0.081 ** (0.09)
cons	6.034 *** (0.73)	cons	6.668 ** (0.75)
R-square	0.7023	R-square	0.7161
N (number of observed values)	450	N (number of observed values)	450
F	96.11	F	88.75

Note: *** and ** denote statistical significance at 1% and 5% level respectively; Fe = Fixed effects.

**Table 7 ijerph-17-07798-t007:** Incremental robustness test results.

Dependent Variable: Increment of Industrial Solid Waste Emission Intensity		
Formal Environmental Regulation		Informal Environmental Regulation
Variable	Fe	Fe
lnfor	−0.166 *** (−0.04)	/
lninfor	/	−0.081 *** (−0.05)
cons	1.255 *** (0.32)	1.402 *** (0.32)
R-square	0.5634	0.5926
N(Number of observed values)	420	420
F	66.33	72.47

Note: *** denotes statistical significance at 1% level; Fe = Fixed effects.

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
