# Peer review of "Effect of Environmental Regulation on Industrial Solid Waste Pollution in China: From the Perspective of Formal Environmental Regulation and Informal Environmental Regulation"

_ijerph, 2020, doi:10.3390/ijerph17217798_

Round 1
Reviewer 1 Report
This paper studied how formal and informal environmental regulation affect the industrial solid waste emission reduction in China. It is a traditional topic. The paper tried to add the effect of informal environmental regulation to the existing studies and paid attention to the threshold effect of the GDP. Some questions that I concerned are as follows. (1)variable definition. In Table 2, the Industrial solid waste emission intensity(ISEI) is the dependent variable, but in Table 4 and 5, the dependent variable is solid waste intensity (SWI). ISEI is SWI? Informal regulation(IR) is defined with EWM by three variables. Authors should explain why they select these three variables in detail. The employee’s income and education level that comes from the interior of organizations does not look like the regulating factors. The control variables have been defined in Table 2 and all have variable names, but why do the x3-x7 be used in Table 4, instead of variable names? (2)Authors should have the descriptive statistics of all variables. If possible, correlation analysis should be done. (3) Authors should explain the reasons that they select the threshold regression instead of linear regression. Because when the threshold regression is used, it defaults that there is a nonlinear relation. (4) In Table 5, the variable actint is an important variable, which is defined as “the interaction of formal and informal environmental regulations” in the paper. But it is no clear that how the actint is calculated in the paper, so authors should show the calculation process in detail. (5) The paper studied the relation between environmental regulation and solid waste emission reduction at the same time point, however, both formal and informal environmental regulation have the time lag effect. So lagging the dependent variable by one year may be a good choice. (6) Because the GDP varies by province, so the province fixed effect and year fixed effect should be considered in the regressions. (7)Authors should consider the marginal contribution to the existing literature. In province level, there is no necessity to study the relation between environmental regulation and solid waste emission reduction. Instead, it is more necessary to study the relation between environmental regulation and solid waste emission reduction in firm level.Author Response
Response to Reviewer 1 Comments
Thank you very much for your attention and the evaluation and comments on our paper. Your comments are of great help in revising our paper. We have revised the manuscript according to your kind advices and detailed suggestions. Please find the responses to your comments below:
Point 1: variable definition. In Table 2, the Industrial solid waste emission intensity(ISEI) is the dependent variable, but in Table 4 and 5, the dependent variable is solid waste intensity (SWI). ISEI is SWI?
Informal regulation(IR) is defined with EWM by three variables. Authors should explain why they select these three variables in detail. The employee’s income and education level that comes from the interior of organizations does not look like the regulating factors.
The control variables have been defined in Table 2 and all have variable names, but why do the x3-x7 be used in Table 4, instead of variable names? 

Response 1: Thanks for the kind suggestion.
(1)The dependent variable is ISEI, and the changes have been made.
(2)The variables defining informal regulation are explained in detail in “3.1.2. Variables”. The definition is based on existing research.
(3)The control variables are defined by variable names.
Point 2: Authors should have the descriptive statistics of all variables. If possible, correlation analysis should be done.
Response 2: Thanks for the kind suggestion.
(1)The descriptive statistics of all variables is added in “4.1. Panel threshold regression analysis”.
(2)According to the theoretical analysis and the conclusions of existing studies, the relationship between the variables selected in this paper is relatively clear, so we think it is not necessary to do correlation analysis before regression analysis. If you still have different opinions, due to time limit, we will revise it in the next round of review.
Point 3:. Authors should explain the reasons that they select the threshold regression instead of linear regression. Because when the threshold regression is used, it defaults that there is a nonlinear relation.
Response 3: Thanks for the kind suggestion.
(1) The explaination of the reasons we select the threshold regression is added in the first paragraph of “3.2. Methods”.
Point 4:. In Table 5, the variable actint is an important variable, which is defined as “the interaction of formal and informal environmental regulations” in the paper. But it is no clear that how the actint is calculated in the paper, so authors should show the calculation process in detail.
Response 4: Thanks for the kind suggestion.
- The detail of how to calculate “actint” please see reference [41]. The detailed revision can be found in the paragraph before Table 6.
Point 5:. The paper studied the relation between environmental regulation and solid waste emission reduction at the same time point, however, both formal and informal environmental regulation have the time lag effect. So lagging the dependent variable by one year may be a good choice.
Response 5: Thanks for the kind suggestion.
(1) Hysteresis is used in the robustness test of this paper. It can be seen from the results that the result of the original threshold environmental regulation is stable. Referring to the existing studies, it is also acceptable not to use lag data for regression. We used a static panel model. So we didn't change this point. However, this suggestion is very important. We have found the possible deficiencies in our study. Due to time constraints, we will make improvements in future studies.
Point 6:. Because the GDP varies by province, so the province fixed effect and year fixed effect should be considered in the regressions.
Response 6: Thanks for the kind suggestion.
(1)Threshold regression is a fixed effect of balancing panel data. In principle, individual effects have been included in the threshold model. Referring to existing research, such as reference [42], there is no need to add the province fixed effect and year fixed effect in the regressions.
Point 7:. Authors should consider the marginal contribution to the existing literature. In province level, there is no necessity to study the relation between environmental regulation and solid waste emission reduction. Instead, it is more necessary to study the relation between environmental regulation and solid waste emission reduction in firm level.
Response 7: Thanks for the kind suggestion.
(1)The marginal contribution of this article is added by the end of “2. Theoretical Background and Hypotheses”. Through literature research, we found that the existing literature is still very lack of research on the impact of environmental regulation on solid waste emission at the provincial level in China.
Reviewer 2 Report
Comments
- Please add a brief presentation of structure of the paper at the end of Introduction section.
- The two Hypotheses H1 and H2 must be together and must are creating a sub-section of the section Materials and Methods (possibly the first sub-section of this section).
- The rest text of sub-sections 2.1.1. and 2.1.2 are the Theoretical Background and they do not own to the section of Materials and Methods. It is better to create a new section under the title Theoretical Background before the section Materials and Methods.
-After the Results section, I suggest authors to create a section of Discussion. Either in Results, either in Conclusions section there are no exist parts that covers totally the content of a Discussion, so manuscript do not provide sufficient discussion and there is no any mention about the originality of the research.
-Are missing the analytical comparison of study' results with the results of previous or similar studies (international oriented).
- In the same section it is necessary to be indicating the originality, the impact from research results to the society, policies and the science (contribution).
- I suggest also to the authors to be add the research limitations and the further research that could be follow on this object.
- The Conclusions section is simple a repetition of the results. So, they do not add nothing. I suggest to Authors to give at the end of the manuscript generalized conclusions together with useful policy proposals or practical recommendations.
- The authors should also rework their manuscript properly correcting some grammatical and syntax errors (language editing).
Author Response
Response to Reviewer 2 Comments
Thank you very much for your attention and the evaluation and comments on our paper. Your comments are of great help in revising our paper. We have revised the manuscript according to your kind advices and detailed suggestions. Please find the responses to your comments below:
Point 1: Please add a brief presentation of structure of the paper at the end of Introduction section. . 

Response 1: Thanks for the kind suggestion.
A brief presentation of structure of the paper is added at the end of Introduction section.
Point 2: The two Hypotheses H1 and H2 must be together and must are creating a sub-section of the section Materials and Methods (possibly the first sub-section of this section). 

Response 2: Thanks for the kind suggestion.
Referring to existing research, it is also possible to discuss Hypothesis 1 and Hypothesis 2 separately, because different hypotheses are preceded by different literature analyses. We have adjusted the structure of the article.
Point 3: The rest text of sub-sections 2.1.1. and 2.1.2 are the Theoretical Background and they do not own to the section of Materials and Methods. It is better to create a new section under the title Theoretical Background before the section Materials and Methods.

Response 3: Thanks for the kind suggestion.
We have adjusted the structure of the article.
Point 4: After the Results section, I suggest authors to create a section of Discussion. Either in Results, either in Conclusions section there are no exist parts that covers totally the content of a Discussion, so manuscript do not provide sufficient discussion and there is no any mention about the originality of the research. 

Response 4: Thanks for the kind suggestion.
A section of Discussion is added.
The marginal contribution of this article is added by the end of “2. Theoretical Background and Hypotheses”.
Point 5: Are missing the analytical comparison of study' results with the results of previous or similar studies (international oriented). 

Response 5: Thanks for the kind suggestion.
The comparison of study' results with the results of previous or similar studies can be found in the section of discussion.
Point 6: In the same section it is necessary to be indicating the originality, the impact from research results to the society, policies and the science (contribution). 

Response 6: Thanks for the kind suggestion.
The marginal contribution of this article is added by the end of “2. Theoretical Background and Hypotheses”.
Point 7: I suggest also to the authors to be add the research limitations and the further research that could be follow on this object. 

Response 7: Thanks for the kind suggestion.
The research limitations and the further research can be found in the section of discussion.
Point 8: The Conclusions section is simple a repetition of the results. So, they do not add nothing. I suggest to Authors to give at the end of the manuscript generalized conclusions together with useful policy proposals or practical recommendations. 

Response 8: Thanks for the kind suggestion.
We have modified the conclusions section with useful policy proposals or practical recommendations.
Point 9: The authors should also rework their manuscript properly correcting some grammatical and syntax errors (language editing). 

Response 9: Thanks for the kind suggestion.
Language editing is done.
Reviewer 3 Report
This study is a valuable paper that quantitatively analyzes the effects of environmental regulation of solid waste in China.
However, the following points should be considered to aid the reader's understanding.
-Although the results of the analysis are expressed in a table, there are parts of the paper where the interpretation of the results is not explained. For example, what exactly is meant by X3-X7 in Table 4?
-(L356)It is said that there is a tendency to have a "U" curve, but please consider how to express this in a way that conveys this tendency, for example by using a figure.
Author Response
Response to Reviewer 3 Comments
Thank you very much for your attention and the evaluation and comments on our paper. Your comments are of great help in revising our paper. We have revised the manuscript according to your kind advices and detailed suggestions. Please find the responses to your comments below:
Point 1: Although the results of the analysis are expressed in a table, there are parts of the paper where the interpretation of the results is not explained. For example, what exactly is meant by X3-X7 in Table 4? 

Response 1: Thanks for the kind suggestion.
We changed X3-X7 by the name of the variables. As for the interpretation of the results, we only focus on the conclusions that are closely related to the research objectives.
Point 2: It is said that there is a tendency to have a "U" curve, but please consider how to express this in a way that conveys this tendency, for example by using a figure. 

Response 2: Thanks for the kind suggestion.
From the positive and negative regression coefficients on two sides of the formal and informal environmental regulation threshold, it can be considered that the impact of environmental regulation on pollution emission intensity presents an inverted u-shaped curve relationship. Based on the existing similar studies, it is not necessary to draw the curve. However, this suggestion is very important. We have found the possible deficiencies in our study. Due to time constraints, we will make improvements in future studies.
Round 2
Reviewer 1 Report
Some minor revisions are needed in this paper.
(1)In the table 3, it can be seen that the difference between the max and min value of continuous variables are very big. It is suggested that all continuous variables be winsorized and rerun the regressions.
(2) It is suggested that the marginal contributions be moved to the section 1 Introduction.
Author Response
Response to Reviewer 1 Comments
Thank you very much for your attention and the evaluation and comments on our paper. Your comments are of great help in revising our paper. We have revised the manuscript according to your kind advices and detailed suggestions. Please find the responses to your comments below:
Point 1: In the table 3, it can be seen that the difference between the max and min value of continuous variables are very big. It is suggested that all continuous variables be winsorized and rerun the regressions.

Response 1: Thanks for the kind suggestion.
This is a good advice. Actually, we had already winsorized the variables before doing the regression, but we didn't write this step into the manuscript before. Now we add the description of this process in the paragraph before Table 3.
Point 2: It is suggested that the marginal contributions be moved to the section 1 Introduction.
Response 2: Thanks for the kind suggestion.
We have moved the marginal contributions to the section 1 Introduction.
Reviewer 2 Report
Please avoid repetitions (in the first round they were limited than in revised edition) and a final check for grammar and syntax mistakes is necessary too.
